# Leaf Membrane Stability under High Temperatures as an Indicator of Heat Tolerance in Potatoes and Genome-Wide Association Studies to Understand the Underlying Genetics

**DOI:** 10.3390/plants13162175

**Published:** 2024-08-06

**Authors:** Amaka M. Ifeduba, Shuyang Zhen, Jeewan Pandey, M. Isabel Vales

**Affiliations:** Department of Horticultural Sciences, Texas A&M University, College Station, TX 77843-2133, USA; shuyang.zhen@tamu.edu (S.Z.); yourjeewan@tamu.edu (J.P.)

**Keywords:** heat stress, heat shock proteins, *Solanum tuberosum*, relative electrolyte conductivity, abiotic stress

## Abstract

High temperatures during the crop growing season are becoming more frequent and unpredictable, resulting in reduced crop productivity and quality. Heat stress disrupts plant metabolic processes that affect cell membrane composition and integrity. Cell membrane permeability, ion leakage, and heat shock proteins have been evaluated to screen for heat tolerance in plants. In potatoes, it is unclear whether leaf membrane stability under heat stress is correlated with underground tuber productivity and quality. The main goal of this study was to evaluate if leaf membrane relative electrolyte conductivity (REC) under high temperatures could be used to identify heat-tolerant potato genotypes. Electrolyte leakage assays, correlation estimations, and genome-wide association studies were carried out in 215 genotypes. Expression levels of small heat shock protein 18 (sHSP18) were evaluated in the heat-sensitive potato variety Russet Burbank and compared with those of the heat-tolerant variety Vanguard Russet using Western blotting. Significant differences were observed among genotypes for leaf membrane REC under extreme heat (50°C); REC values ranged from 47.0–99.5%. Leaf membrane REC was positively correlated with tuber external and internal defects and negatively correlated with yield. REC was negatively correlated with the content of several tuber minerals, such as nitrogen, magnesium, and manganese. Eleven quantitative trait loci (QTLs) were identified for leaf membrane REC, explaining up to 13.8% of the phenotypic variance. Gene annotation in QTL areas indicated associations with genes controlling membrane solute transport and plant responses to abiotic stresses. Vanguard Russet had lower leaf REC and higher expression of sHSP18 under high-temperature stress. Our findings indicate that leaf membrane REC under high temperatures can be used as an indicator of potato heat tolerance.

## 1. Introduction

Exposure of plants to temperatures higher than their optimum for growth and development negatively impacts many cellular metabolic and physiological processes, such as photosynthesis and respiration, and affects membrane stability, protein integrity, and expression [1,2,3]. High-temperature stress damages cell membranes, causing electrolyte leakage, the degradation of nucleic acids, and the denaturation of proteins [4]. These changes due to extreme heat indicate that cellular membranes are involved in heat-induced cellular damage in many plant species [5,6]. Cell membranes are mainly composed of proteins and fatty acids. Fatty acids can become more unsaturated in response to high-temperature stress [7]. These changes contribute to the adjustment of the physical characteristics of the membranes, enabling them to function effectively across different environmental conditions. If environmental temperatures exceed the normal limits of plant growth and development, cell membranes undergo major structural changes [8]. These structural disturbances include phase separation of the membrane constituents associated with loss of selective permeability, ion leakage, and changes in protein expression and transport processes. Many transmembrane proteins function as gateways to permit the transport of specific substances across the membrane. They frequently undergo significant conformational changes to move a substance through the membrane and are disrupted by high-temperature stress [9].

Electrolyte or ion leakage assays have been used as an indirect method to measure heat tolerance in diverse plant species. For example, cotton [10,11], tomato [12], wheat [13,14,15], chili pepper [16], and potatoes [2,17]. The ion leakage test determines the degree of cell membrane injury caused by heat stress based on electrolyte leakage from the cells. It is a relatively simple, rapid, and repeatable technique that requires inexpensive equipment and may be suitable for screening many samples. It offers a non-destructive assessment requiring only a few leaf samples and quantitative results that are reproducible. It also provides insights into plant responses to heat stress, serving as a quick screening method to discriminate between heat-tolerant and heat-sensitive varieties. The method is also employed to quantify cell membrane damage caused by other abiotic stressors such as low temperatures [18], as well as responses to biotic stresses like wheat leaf rust [19] or rice sheath blight [20].

Potato (*Solanum tuberosum* L.) is the most important vegetable crop in the world. It is grown globally in 160 countries [21,22] under various climatic conditions. However, potato is a cool-season crop with optimal growth temperatures during the day of 20–25°C and night temperatures 10 to <20°C [23]. High temperatures during the growing season result in reduced marketable yields; reduced specific gravity (starch and solids); an increase in external defects (knobs, gemmations, growth cracks, linked tubers, and heat sprouts), and internal defects (brown center, hollow heart, vascular discoloration, and internal heat necrosis); quality deterioration (for chips and French fries); increase in reducing sugars; and reduction in the dormancy period [24]. Night temperatures above 20°C result in significant yield reduction, and for some varieties, growing at night temperatures exceeding 25°C may result in a total crop loss. Some potato cultivars such as Red LaSoda, Bravo, Kufri Surya, Coastal Chip, and Tacna have been bred/selected for enhanced heat tolerance for warmer climates. Traditional breeding based on selection under high field temperatures is one of the most frequently used practices for developing heat-tolerant varieties [25,26,27,28,29,30]. However, it is riddled with many challenges and limitations. Field trials are affected by unpredictable weather, variability in soil type, differences in soil fertilization and moisture distribution, incidences of pests and diseases, etc., and can usually accommodate only a limited number of genotypes [23,31]. A few studies have had limited success in screening for heat-tolerant varieties using *in vitro* techniques like bioreactors or screening under controlled growth chambers or greenhouse conditions [32]. Kim et al. [27] developed a rapid *in vitro* screening method for heat tolerance screening based on the tuberization of plants grown from true seeds to eliminate the need for transplanting shoot families. They showed limited success in using this technique to identify heat-tolerant families. Gautam et al. [24] studied the genetic basis of traits linked to heat stress traits, but the genetics are complex, and the percentage of the phenotypic variance explained was low. There was more success in identifying quantitative trait loci (QTLs) when individual tuber defects were dissected. Identifying traits and molecular markers linked to heat tolerance and developing fast screening methods (direct or indirect) to identify heat-tolerant clones are highly desirable.

Electrolyte leakage assay has been employed as a screening method for potato heat tolerance. Arvin et al. [17] used this method to screen the tolerance level of potato cultivars and wild species to several abiotic stresses, including heat, and showed that correlations among types of stress tolerance were significant for all stresses except for heat and drought. Savic et al. [2] also used this method to assess heat tolerance in nine commercial potato cultivars. They reported that Laura was tolerant, while Arnova, Agria, and Carrera were relatively heat-susceptible. Additional studies have been conducted using electrolyte assays to evaluate potato heat stress [33,34,35]. Determining whether leaf membrane stability is connected to producing high yields of marketable tubers with few external and internal defects is of great interest because it would facilitate the rapid screening of heat-tolerant potato varieties to guide breeding efforts.

In addition to adjusting cell membrane fluidity, plants have evolved other adaptive mechanisms to mitigate the detrimental impacts of high-temperature stress [36]. Synthesizing small heat shock proteins (sHSPs) is an example of such a mechanism [37,38]. sHSPs act as molecular chaperones that promote the refolding of heat-denatured proteins or form complexes with them to prevent irreversible thermal aggregation [39,40,41,42]. In addition, several studies have indicated that sHSPs are essential in maintaining membrane integrity under stress conditions [43]. A strong correlation between the synthesis of sHSPs and the development of heat tolerance has been reported in several crops, such as soybean [44], sorghum [45], apple [46], rice [47], tomato [48], barley [49], and durum wheat [50]. Savic et al. [2] found a higher synthesis of HSP18 and HSP21 in heat-tolerant potato genotypes than in heat-sensitive genotypes.

Genome-wide association study (GWAS) is a powerful strategy for identifying molecular markers linked to complex traits [51]. This approach identifies quantitative trait loci (QTLs)/genes by examining marker-trait associations based on the strength of linkage disequilibrium (LD) between the markers and functional polymorphisms throughout diverse germplasm [52]. The GWASpoly method, introduced by Rosyara et al. [53], incorporates the kinship matrix (K) in potatoes and considers allelic dosage.Yang et al. [54] proposed the leave-one-chromosome-out (LOCO) method to enhance computational efficiency. The LOCO method calculates a separate covariance matrix for each chromosome based on markers from all other chromosomes. GWASs have been used in many tetraploid potato studies Vos et al. [55] used GWAS to identify SNPs associated with glycoalkaloid content; Pandey et al. [56] detected QTLs associated with tuber shape, eye depth, flesh color, and other tuber traits Zia et al. [57] performed a GWAS for various morpho-agronomic traits. Kaiser et al. [58] identified genetic features associated with common scab resistance; and Klaassen et al. [59] evaluated protein content. Therefore, GWAS offers the potential to dissect the genetics of complex traits using molecular markers.

The Texas A&M University Potato Breeding Program has over 200 advanced clones selected over 40 years and maintained *in vitro* [56,60]. Over many years, the clones have been propagated and evaluated under field conditions in Texas, USA, a region prone to heat stress. They have been used for GWAS and genomic selection (GS) studies of several traits, such as tuber amino acid contents [61], minerals [62], and external and internal tuber defects [24]. Since these clones are widely diverse, this panel offers the opportunity to explore the genetic basis of leaf membrane integrity.

The main goal of this study was to evaluate if leaf cell membrane integrity could be used to identify heat-tolerant potato varieties. The specific objectives were to (a) assess potato leaf membrane REC variation in a panel of 215 genotypes exposed to heat; (b) correlate leaf REC with external and internal tubers defects, tuber minerals, and yield; (c) identify quantitative trait loci (QTLs)/genes responsible for leaf membrane REC; and (d) assess the differences in expression levels of the small heat shock protein 18 (sHSP18) between the heat-sensitive clone Russet Burbank and the heat-tolerant variety Vanguard Russet.

## 2. Results

### 2.1. Extent of Electrolyte Leakage Observed

**Initial study**: The results from the initial experiment using known heat-tolerant and heat-sensitive genotypes showed an increase in leaf membrane ion leakage (relative electrolyte conductivity, REC) as the temperatures increased from 30 to 70°C for all five genotypes (Figure 1A). Significant differences existed between the heat-tolerant (low ion leakage) and heat-sensitive (high ion leakage) potato genotypes at most temperatures, with Sierra Gold^TM^ having intermediate ion leakage levels. Russet Burbank and Atlantic (heat-sensitive) showed greater leaf membrane REC with increasing temperature than other genotypes, reaching 92.4% and 91.4%, respectively, at 70°C. Sierra Gold^TM^ (heat-tolerant) had lower REC at 70°C (79.3%) than the heat-sensitive clones but more than Reveille Russet and Vanguard Russet (heat-tolerant), with REC of 61.8% and 66.7% at 70°C, respectively (Figure 1A). The temperature at which 50% electrolyte leakage occurred (LT50) was calculated to indicate the extent of cell membrane injury when exposed to heat. Significant differences in LT50 between the genotypes were found. Vanguard Russet had the highest LT50 (64.8°C), whereas Russet Burbank had the lowest LT50 (45.0°C) (Figure 1B).

**Main study**: In the main study, Russet Burbank Atlantic had high REC (79.8% and 82.5%, respectively), significantly higher than Sierra Gold^TM^ (71.4%) (intermediate REC) and Vanguard Russet and Reveille Russet (63.15% and 63.41%, respectively) (low REC) (Figure 2). There was a continuous distribution for relatively electrolyte conductivity for the 215 clones screened at a temperature of 50°C ranging from 47 to 99.5% (Figure 3). The distribution was Gaussian based on Shapiro–Wilk’s test, where W = 0.9897 and Prob < W = 0.0882. A few clones like AOTX96216-2Ru, TX17797s-11Y/Y, NDTX060700C-1W, TXYG-57, and Tacna showed very high REC, greater than 90% REC, while a few like TX12474-1P/R, ATX13134-3W/Y, COTX08365-1P/P, Dubloon and COTX17304-1Ru showed low REC of less than 55% (Appendix A). Of the 62 genotypes in the Russet group, 15 had REC higher than Russet Burbank, 7 had REC lower than Vanguard Russet, and 9 lower than Reveille Russet. Sierra Gold^TM^ is positioned in the middle. Of the 41 genotypes in the chippers group, only 9 had REC higher than Atlantic. Therefore, the heat-sensitive and tolerant varieties from the initial study (Russet Burbank, Atlantic, Sierra Gold^TM^, Reveille Russet, and Vanguard Russet) showed a similar trend (but higher values in the main study since the incubation took place for 4 h—instead of 30 min for the initial experiment). The repeatability estimate was high (0.96). Based on the phenotypic distribution of the individual market classes, the red group had fewer extreme classes (no genotypes with <60% and >90% REC); the russets, yellows, and chippers had members with >90% REC; and the russets, yellows, and purples had members with <60% REC (Figure 4) when all market class members were considered. On average, chips had significantly higher REC than russets, purples, and yellows, but chips and reds were not significantly different for REC (Appendix A).

### 2.2. REC Significantly Correlated with Tuber Yields, Defects, and Mineral Content

The correlation analysis showed a significant negative correlation (−0.64) between the Z-score weighted multi-index selection for overall performance and leaf membrane REC and a significant positive correlation between leaf membrane REC and external (0.40) and internal (0.34) tuber defects (Table 1). When tuber defects were dissected by type, a significant positive correlation was found between leaf membrane REC and percent tubers with knobs (0.32) and percent tubers with internal heat necrosis (0.27) (Table 1). Also, the correlation of REC with tuber minerals indicated a significant negative correlation with nitrogen, N (−0.21); magnesium, Mg (−0.25); manganese, Mn (−0.22); and sulfur, S (−0.20) and a significant negative correlation (−0.25) between the Z-score weighted multi-index selection for total tuber mineral and leaf membrane REC (Table 2).

### 2.3. Genome-Wide Associations

A total of 11 significant QTLs were identified by genome-wide association studies using GWASpoly for leaf membrane REC in 6 of the 12 potato chromosomes (1, 3, 4, 5, 7, and 11). The QTLs explained varying percentages of phenotypic variance, from less than 1 to 13.8% (Figure 5, Table 3). Several key genes involved in abiotic stress response and membrane stability were in areas where QTLs for REC were identified. A candidate gene for the QTL detected on chromosome 7 is a Calmodulin-binding protein, which plays a vital role in Ca^+2^-mediated signaling [63]. A potential gene for the QTL detected on chromosome 3 is the F-box family proteins involved in vegetative and reproductive growth [64]. For the QTL detected on chromosome 1, the likely candidate gene is the Clathrin assembly protein important for vesicle trafficking and membrane stability [65]. Additionally, several isoforms of heat shock proteins were detected near QTLs on multiple chromosomes and are crucial for membrane quality control under stress conditions [43]. The location of significant QTLs on different chromosomes was visualized using chromoMap package (Version 4.1.1) (Figure 6).

### 2.4. Heat Shock Protein 18

Under normal temperatures, HSP18 expression was not detected in Vanguard Russet and Russet Burbank, which was expected because HSPs are rarely detected when plants grow under optimal conditions. However, under high-temperature stress, HSP18 had significantly higher expression after 18 h of heat stress (35°C) in the heat-tolerant genotype Vanguard Russet (almost threefold higher) than the heat-sensitive Russet Burbank (Figure 7). Similar results were obtained from two independent experiments (Figure 7).

## 3. Discussion

As global temperatures increase, it becomes more important to develop new cultivars that are tolerant to heat stress and/or to identify existing varieties that are tolerant to high-temperature stress. Developing a rapid screening technique for potato heat tolerance is highly desirable to the potato industry as it would guide breeding efforts toward developing heat-tolerant potato varieties. Due to selection under heat stress, some potato cultivars released by the Texas A&M Potato Breeding Program and advanced clones have been characterized as heat-tolerant (high yields of marketable tubers with few external and internal defects) since selection took place under high temperatures during the growing season [24]. In the present study, the effects of heat stress on leaf membrane permeability were evaluated by incubating leaf tissue at high temperatures. It is unclear if leaf membrane REC could be used as a predictor of heat tolerance. Based on our initial study, we determined that heat stress increased the membrane permeability significantly as temperature increased in Atlantic and Russet Burbank, the heat-susceptible genotypes, and less in Vanguard Russet and Reveille Russet, the most heat-tolerant genotypes. Sierra Gold^TM^, with moderate tolerance to heat stress, had a REC average between the two (Figure 1A). We calculated the LT50 among the five genotypes and observed significantly higher values in Vanguard Russet and Reveille Russet, the most heat-tolerant genotypes (Figure 1B). This indicates that the level of cell membrane injury under high-temperature stress was higher in Russet Burbank and Atlantic than in Vanguard Russet and Reveille Russet. From the sigmoid curve, we observed a clear distinction among the five genotypes in their relative electrolyte leakage at 50°C (Figure 1A). This observation informed our selection of 50°C as the best temperature to phenotype the 215 genotypes used in the main studies (Figure 3). In a similar study, Ahn et al. [33] showed that the heat-tolerant Norchip potato variety had lower electrolyte leakage when compared to heat-sensitive Atlantic and Russet Burbank. Savic et al. [2] also reported that the potato variety Laura was tolerant, while Arnova, Agria, and Carrera were relatively heat-susceptible based on electrolyte leakage assay. However, it is important to note that the overall thermotolerance of potatoes is determined by several genetic and environmental factors, and the method described may serve as an initial selection method for a large panel. To be declared heat-tolerant, a variety must produce a high yield of marketable tubers under high-temperature stress conditions.

The phenotypic distribution of the 215 genotypes was normal based on Shapiro–Wilk’s test, indicating quantitative variation and likely the presence of several minor genes involved in explaining the REC trait. Vanguard Russet and Reveille Russet showed the same pattern as indicated in the initial study, with significantly lower ion leakage than Atlantic and Russet Burbank (Figure 2 and Figure 3), despite showing different REC magnitudes because of the different incubation times (30 min in the initial study vs. 4 h in the main study). We adopted an incubation time of 30 min in the initial study to develop a sigmoid curve and determine the best temperature where the genotypes showed a clear separation. The high repeatability of leaf membrane REC (0.96) in this study is an indication that the genetic component of the trait is considerable. A strong correlation existed between the percentage of relative electrolyte leakage and heat tolerance-related traits, including total yield and total internal and external defects. The strong negative correlation (−0.64) between the best-performing genotypes (based on weighted multiple index selection) and relative electrolyte conductivity indicated that clones that performed better in the fields also had lower electrolyte leakage. Also, the positive correlation of 0.40 and 0.34 between relative electrolyte conductivity and external and internal tuber defects shows that clones with higher REC also tended to have more internal and external defects (Table 1). When tuber defects were dissected, we identified a significant positive correlation between REC and percent tuber with knobs (0.32) and percent tubers with internal heat necrosis (0.27), which further proves the correlation between REC and heat-tolerance-related traits (Table 1). The positive correlation between REC and tuber defects suggests that high-temperature stress causes electrolyte loss above ground, disrupting cell membrane integrity. This disruption limits the translocation of photoassimilates to the tubers, negatively impacting tuber development and leading to internal and external defects. Additionally, tuber cell membranes may suffer similar damage to leaf membranes, where ion leakage could impair cell division, resulting in knobs and growth cracks, or cause cell death, leading to internal heat necrosis. The correlation of REC with tuber minerals showed negative correlations for sodium, magnesium, manganese, and sulfur, as well as the total minerals combined (ZWMIS) (Table 2). This indicates that genotypes with higher tuber minerals tended to have lower electrolyte leakage and vice versa. We observed that the chipper (NDTX050169s-1R) with the lowest REC (55.63%) was the same genotype that Pandey et al. [62] reported as having the highest mineral content. Therefore, we infer that under high-temperature stress, there is an increasing loss of electrolytes from the above-ground parts, which limits its translocation to the tubers, thus resulting in lower tuber minerals under these conditions.

The genotypes evaluated showed varying REC, which could be for several reasons. White LaSoda (included in the yellow group but has some pink streaks and white flesh) is a white mutant of Red LaSoda. The two genotypes showed similar low REC values (63.73% and 67.14%). Previous studies have shown that Red LaSoda tolerates heat stress well in Texas and produces high yields of marketable tubers with few external and internal defects [24]. This could be the reason why both genotypes showed a relatively lower REC. Of all the strains of Russet Norkotah evaluated in this study, Russet Norkotah 223, 296, and 278 are the most popular. They have been reported to be more vigorous and have later maturity than other strains [66]. This could explain why they had relatively lower REC (53.72%, 68.87%, and 67.33%) than other less vigorous strains, such as 102 and 112, which had higher REC similar to the original Russet Norkotah (Appendix A). Tacna has been described as heat-tolerant [26], but several years of our field trials have indicated that it has low marketable yield when grown under heat-stress conditions in Texas. Therefore, it was unsurprising that Tacna had a high REC (90.08%) in the current studies. Our results indicate that leaf electrolyte leakage assay may serve as a quick and indirect assessment of potato heat tolerance at the tuber level.

Understanding the genetic basis of REC can also help to select for heat tolerance indirectly. This was done by conducting genome-wide association studies between molecular markers and the leaf membrane REC phenotypes. In the present study, a panel of 215 tetraploid potato clones was genotyped with a high-density SNP marker array and phenotyped for relative electrolyte leakage as described using GWASpoly. Most genomic regions detected have been uncovered for the first time (Figure 5, Table 3). The significant QTL detected on chromosome 7 (SNP at peak: solcap_snp_c1_7381 at a position 42.6 Mb) explained about 13.8% of the phenotypic variance of the simplex-dominant-reference allele (1-dom-ref) model. The likely candidate gene for this QTL is the Calmodulin-binding protein (CaMs), the key player in plants’ Ca^+2^ mediated abiotic stress signaling cascade. They sense the altered Ca^+2^ concentrations in the cell cytosol and are actively involved in signal perception and transmission under high-stress conditions [63]. An isoform of this gene, PGSC0003DMG400032792, located in position 43.4 Mb, is less than 0.8 Mb from the target SNP at chromosome 7. We also found several isoforms of the CaMs close (less than 2 MB) to the target SNP in chromosome 3. Tawfik et al. [67] reported that supplemental treatment of potato leaves with calcium resulted in better membrane stability and lower ion leakage. This indicates that calcium is critical in mitigating heat stress effects on potato plant growth.

The leaf membrane REC QTL on chromosome 3 explained about 8.3% of the phenotypic variation in the 1-dom-ref model. The SNP at the peak of the QTL on chromosome 3 is PotVar0056569. One of the candidate genes of potential interest is located at 50. 9 Mb, namely the F-box family protein, which is one of the super protein families in eukaryotic cells involved in vegetative and reproduction growth [64]. We found several isoforms of the F-box family protein close (less than 0.1 Mb) to the peaks of QTLs on chromosomes 3, 5, and 11. The QTL detected on chromosome 1 (SNP at peak: PotVar0071937 at 1.2 Mb) explained 2.6% of the phenotypic variations (additive model). A likely candidate gene for this QTL is the Clathrin assembly protein, which functions in the coating of intracellular vesicles that transport cargo between membrane-bound compartments. Clathrin-coated endocytic vesicles are produced by a complex modular protein machinery that transiently assembles on the plasma membrane. Forces arising within the membrane during deformation counteract forces generated by the endocytic protein modules. Physical parameters such as membrane tension and rigidity control the dynamics of clathrin-mediated endocytosis [65]. Two isoforms of this gene (PGSC0003DMG400041999 and PGSC0003DMG400022769) are located about 0.7 and 0.8 Mb from the QTL peak on chromosome 1 and close to the target QTL peak on chromosome 4 represented by SNPs (PGSC0003DMG400029522 and PGSC0003DMG400029445).

We detected several isoforms of the heat shock binding protein gene close to (less than 0.1 Mb) from the QTL peaks of chromosomes 3, 4, 5, and 7. This shows that this gene possibly plays a critical role in structural changes that occur in leaf membrane when exposed to heat stress and confirms our analytical finding using Western blotting to show that the expression of heat shock protein is increased during high-temperature stress and is better expressed by the tolerant varieties, which leads to better stability of their membranes. Other studies have indicated that heat shock proteins play an essential role in membrane quality control and contribute to maintaining membrane integrity under stress conditions [43].

Generally, most candidate genes corresponded to proteins with enzyme activity like ATPase and protein tyrosine phosphatase. ATPase is critical in transporting solutes throughout the cell membrane [68], while protein tyrosine phosphatase plays a critical role as a regulator of abiotic stress tolerance in plants [69]. ElBasyoni et al. [5] used electrolyte leakage assay and genome-wide association studies in a wide accession of wheat and detected significant QTLs associated with cell membrane integrity in chromosomes 2, 3, 4, 6, and 7. Their study, like ours, identified potential new genomic regions associated with cell membrane integrity and reported that genotypes with better membrane stability based on electrolyte leakage were more productive under preliminary field stress conditions. Overall, the gene annotation for most QTLs we detected indicated that they are involved in the transport of solutes through the cell membrane or play an important role in abiotic stress tolerance. However, our study is one of the few that have employed electrolyte leakage assay to a wide potato panel and incorporated genome-wide association studies to understand the underlying genetics of heat-induced membrane damage and estimate the correlation between REC and field performance, as well as REC and quality.

For the expression levels of sHSP18 using immunoblot analysis (Western blotting), we observed that the heat-tolerant Vanguard Russet had almost a threefold expression more than the heat-sensitive Russet Burbank based on the size of the bandwidth of the protein (Figure 7. Using a similar technique, Savic et al. [2] compared the expression of sHSPs in some potato varieties and reported that heat-tolerant Laura accumulated higher levels of sHSPs than heat-sensitive Liseta and Agria. Also, Ahn et al. [33] observed differential expression of HSP in different potato cultivars. According to them, the expression levels of the 18 kDa sHSP in the heat-sensitive Russet Burbank and Atlantic cultivars increased quickly after heat shock, and its accumulation was maximal between four to eight hours and then decreased drastically with continuous heat shock. Meanwhile, in heat-tolerant ‘Norchip’ and ‘Désirée,’ the 18 kDa sHSP accumulated gradually and increased continuously up to 24 h before decreasing. This finding agrees with our present study, i.e., 18 h after heat stress, the accumulation of sHSP18 in the heat-sensitive Russet Burbank was already declining, while the heat-tolerant Vanguard Russet still had a strong expression. Trapero-Mozos et al. [70] using a bioinformatics-based approach, showed that the expression of several HSP20 genes (*STHsp20s*) was upregulated under heat stress. Based on differential expression observed in heat-tolerant and heat-sensitive cultivars, HSPs have been proposed as a potential heat-tolerance marker for detecting heat-tolerant potato genotypes [71,72]. The Western blotting procedure assesses protein expression levels, which can complement genotypic markers identified through molecular techniques like PCR-based markers or sequencing. By analyzing protein expression patterns in different genotypes, potential markers can be identified and developed to integrate marker-assisted selection using HSPs as a screening tool for potato heat tolerance.

## 4. Materials and Methods

### 4.1. Electrolyte Leakage Assay

#### 4.1.1. Plant Materials and Growing Conditions

**Initial study**: Five potato genotypes were used: Vanguard Russet, Reveille Russet, Sierra Gold^TM^, Russet Burbank, and Atlantic. Previous field and greenhouse experiments have shown that Vanguard Russet, Reveille Russet, and Sierra Gold^TM^ have some inherent tolerance to heat stress, while Russet Burbank and Atlantic are heat-sensitive [24]. Vanguard Russet and Reveille Russet are fresh market russet potato varieties that produce high yields of marketable tubers with few external and internal defects. Sierra Gold^TM^ is a unique potato with russet skin and yellow flesh used mainly for the fresh market. It produces high yields of marketable tubers and few tuber defects under heat-stress conditions. Russet Burbank is a processing russet variety commonly used for French fries (number one in the USA) that produces low yields of marketable tubers and a high percentage of defects (mainly external deformities) under high-temperature conditions. Atlantic is a chipper (number one in the USA) that, despite producing high yields of seemingly marketable tubers, has a high percentage of tubers with internal defects (mainly internal heat necrosis) that make them unmarketable.

Four mini-tubers of each variety (derived from *in vitro* propagation of disease-free tissue culture plantlets grown in soil inside a greenhouse) were planted in four-liter pots (one tuber per pot) containing soil media Pro-Mix BX (Premier Tech, Quakertown, PA, USA) and starter fertilizer Osmocote 14-14-14 (Scotts Miracle-Gro, Marysville, OH, USA). The plants were grown in the greenhouse at an average daily temperature of 25 ± 5.5°C, night temperatures of 18.3 ± 4.8°C, and a photoperiod of approximately 16 h of light and 8 h of darkness. Peter’s fertilizer 20-20-20 (100 ppm Nitrogen) (The Scotts Company, Marysville, OH, USA) was applied weekly starting 30 days after planting (DAP). Plants were watered as needed to avoid drought stress. At each irrigation event, plants were watered to field capacity.

**Main study**: A diverse panel containing 215 tetraploid potato clones was used. The panel contained advanced clones entered into tissue culture by the Texas A&M Potato Variety Breeding Program over several decades, varieties released by the same program, and reference varieties for various market groups. The collection comprised 62 russet, 41 chipping, 31 yellow-skinned, 62 red-skinned, and 19 purple-skinned clones (201 were included in Pandey et al. [60] and 14 additional clones) (Appendix A). Some of the clones were clonal strain selections. As reported earlier, the analysis of population structure and discriminant analysis of principal components displayed three sub-populations [60]. The micropropagation of disease-free tissue culture plantlets in Murashige and Skoog (MS) media (Caisson Laboratories Inc., Smithfield, UT, USA) containing 3% sucrose was carried out to generate eight plantlets of each clone. Fully rooted tissue culture plantlets were transplanted into 32-cell (8 × 4) nested greenhouse growing trays (38 × 28 × 6 cm—height, width, and depth) (Cezoyx, Salt Lake City, UT, USA), acclimated in a greenhouse equipped with an intermittent misting system for two weeks at a day/night temperature of about 25/15°C. Afterward, they were moved to a greenhouse under similar temperature conditions and photoperiods as described in the initial study. Each genotype was replicated four times (four pots/genotype), and the genotypes were planted randomly.

#### 4.1.2. Sample Collection and Heat Treatment

Five leaf discs (10 mm in diameter) were taken from the 3rd or 4th leaf from the top of two-month-old greenhouse plants, avoiding the leaf veins, and placed in glass vials containing 10 mL of deionized water. Three sets of each genotype (four reps each) were prepared for the initial and main studies. The sets consisted of (1) a control set, including samples incubated on an orbital shaker (VWR Scientific Radnor, Radnor, PA, USA) at room temperature, 22 ± 2°C, for four hours; (2) a total ion leakage set, including samples autoclaved for 20 min at 121°C; and (3) a heated set, for which the heat treatment was applied (at a temperature ranging from 30 to 70°C for the initial experiment and 50°C for the main experiment).

For the initial study, the heated samples were transferred to an oscillating water bath (SWBR 27 Shel Lab Bath, Sheldon Manufacturing Inc., Cornelius, OR, USA) set at temperatures ranging from 30 to 70°C with 5°C increments (a total of nine different temperature set points). Samples were incubated for 30 min at each temperature set point to obtain a sigmoid curve. We incubated the samples for 30 min at a wide range of temperature set points in the initial study to develop a temperature response curve and determine the optimal temperature at which the genotypes showed a clear separation. Different samples were used for each temperature point. The temperature was monitored by a CR1000x datalogger (Campbell Scientific Inc., Logan, UT, USA) connected to a computer. Each sample’s electrical conductivity (EC) was measured using an EC meter (Hanna Instruments Inc., Woonsocket, RI, USA). The relative electrical conductivity (REC) at each temperature set point was calculated using the following formula: REC = 100 × [{EC Heated − EC control}]/EC Total.

For the main study, the temperature used for the heated set was 50°C, and leaf membrane REC values were determined after four hours of incubation in the water bath using a Mettler Toledo conductivity meter (SevenDirect SD30, Greifensee, Switzerland) following the method described by Savic et al. [2]. The temperature of 50°C was selected for large-scale leaf membrane REC screening since it was very stringent (based on the initial study) and allowed for the clear differentiation of REC values between clones contrasting for heat tolerance. For the main study, all genotypes were incubated for four hours at 50°C (temperature where optimal separation was observed in the initial study) to obtain the total electrolyte leakage of the samples. Previous authors have employed various incubation times ranging from two to four hours at 37–50°C (Ahn et al. [33]; Savic et al. [2] to obtain the total electrolyte leakage. We chose four hours at 50°C to be more stringent.

#### 4.1.3. Data Analysis for Electrolyte Leakage Assay

For each genotype in the initial experiment, leaf membrane REC vs. temperature was plotted. REC data were fitted to a four-parameter sigmoidal function as y = y_0_ + a/{1 + exp[−(x − x_0_)]/b} using SigmaPlot (version 12.3; Systat, San Jose, CA, USA). The temperature at which each genotype lost 50% of its electrolytes (LT50) was also obtained using SigmaPlot to indicate the extent of cell membrane injury. For the main study, analysis of variance, mean comparisons, phenotypic distribution curve, and Shapiro–Wilk’s test for normality were carried out using JMP pro 17^®^ (SAS Institute, Cary, NC, USA), and the best linear unbiased estimates (BLUEs) was calculated using META-R [73]. Repeatability was calculated by dividing the genotypic variance by the phenotypic variance.

### 4.2. DNA Extraction and Genotyping

Genomic DNA was extracted from 50 to 80 mg of fresh young potato leaves from tissue culture plantlets using the DNeasy Plant Pro Kit (Qiagen, Valencia, CA, USA) for the panel of 215 genotypes (main study). Samples were genotyped using the Infinium 22 K V3 Potato Array on the Illumina iScan (Illumina Inc., San Diego, CA, USA) at Michigan State University. The marker dataset was filtered to retain SNPs with a 90% call rate and a minor allele frequency of at least 0.05, as described in Pandey et al. [60].

### 4.3. Genome-Wide Association Studies and Correlation Analysis

The GWASpoly package (Version 2.12) was used to conduct a genome-wide association study (GWAS) to identify genomic regions associated with membrane integrity [53]. After filtration, 10,707 markers (Appendix A) and the phenotypic data (REC values, Appendix A) were used for the analysis. The leave-one-chromosome-out (LOCO) method was implemented to account for population structure. The Bonferroni test was conducted to control the false discovery rate at 5% significance and establish an LOD threshold. Additive and dominant genetic models were tested. The location of significant QTLs on different chromosomes was visualized with the package chromoMap (Version 4.1.1) [74] (Figure 6). The putative candidate genes in QTL regions were identified by retrieving genes from the potato reference genome of *Solanum tuberosum* group Phureja DM1-3 PGSC v4.03 from the Spud database (http://solanaceae.plantbiology.msu.edu/, accessed on 2 February 2024)

Most of the clones included in this study were evaluated in a separate study for field performance under heat-stress conditions over two years and at two locations. Yield and tuber defects (internal and external), genomic estimated breeding values (GEBVs) [24], and tuber mineral data [62] were combined with leaf membrane REC data (from this study) to calculate Pearson correlation using JMP pro 17^®^ (SAS Institute, Cary, NC, USA).

### 4.4. Plant Materials and Growing Conditions for Heat Shock Proteins

Two cultivars, Vanguard Russet and Russet Burbank, with contrasting response to heat stress (Vanguard Russet is heat-tolerant, and Russet Burbank is heat-sensitive), were used to analyze sHSP18. The plants were laid out in a completely randomized design in two walk-in Conviron BDW80 growth chambers (Winnipeg, MB, Canada). Both chambers were maintained at normal temperature (25/15°C, day/night) for the first 30 days. The photoperiod was set at 16:8 h (light/dark), and the light intensity was set at 600 µmol/m^2^/s. Relative humidity was set at 70% and CO_2_ at 400 ppm (normal atmospheric CO_2_). Mini-tubers of each cultivar were planted in four-liter pots (one tuber per pot) containing soil media Pro-Mix BX (Premier Tech, Quakertown, PA, USA) and starter fertilizer Osmocote 14-14-14 (Scotts Miracle-Gro, Marysville, OH, USA). Plants were watered as needed to avoid drought stress. Each time, plants were watered to field capacity.

At 30 days after planting, the temperature of one growth chamber was switched to heat stress (35°C), while the control plants were maintained at normal temperature (25°C). Samples (leaf tissues) were collected from both chambers 18 h after initiating heat treatment. Two leaf discs (10 mm in diameter) were taken from the 3rd or 4th upper leaf, avoiding the leaf veins, and placed in 1.5 mL microcentrifuge tubes immediately frozen in liquid nitrogen and stored at −80°C until further use. Each genotype was replicated four times.

### 4.5. Heat Shock Protein Assessment and Immunoblot Analysis

Total soluble leaf proteins were extracted using 20 mM Tris + 20 mM B-mecaptoethanol pH-6.8 buffer, with 2 mM ethylenediaminetetraacetic acid (EDTA), 10% glycerol and 1% protease inhibitor cocktail for plant cell and tissue extracts (Sigma Aldrich, St. Louis, MO), and then centrifuged at 14,000× *g* for 5 min at 4°C [2]. Protein concentrations were determined according to Bradford M. [75]. Equal amounts of protein (20 µg) were loaded and separated on 12.5% polyacrylamide gels at 100 volts for 15 min and then for 45 min at 150 volts. The gel thickness and time for electrophoresis were decided after several rounds of optimization. Following gel electrophoresis, the proteins were transferred to a polyvinylidene fluoride (PVDF) membrane (Bio-Rad, Hercules, CA, USA), and the blots were processed as described by Momcilovic et al. [76]. To prevent the unspecific binding of antibodies to the membrane, the blots were probed with polyclonal anti-HSP17.6 (against *Arabidopsis thaliana* HSP17.6 CI recombinant protein) (Agrisera AB, Vännäs, Sweden). The blots were treated with a few drops of SuperSignal^TM^ West Femto Maximum Sensity Substrate (ThermoFisher Inc., Waltham, MA, USA), and visualized using a charged-coupled device (CCD) camera (DXM1200) (Nikon Instruments Inc., Tallahassee, FL, USA). The expression levels of sHSP18 were estimated by determining the bandwidth with ImageJ software (ver. 5.2, Molecular Dynamics, Sunnyvale, CA, USA).

## 5. Conclusions

We observed that heat-tolerant potato genotypes tended to have lower leaf relative electrolyte conductivity (i.e., greater membrane integrity) than heat-sensitive genotypes under high temperatures. The genome-wide associated studies identified 11 QTLs primarily associated with genes responsible for transporting solutes through the cell membrane or for abiotic stress tolerance. We demonstrated the use of genome-wide association studies to identify genomic regions associated with cell membrane stability. Heat stress disrupts the membrane integrity, leading to electrolyte loss and less stable cell membranes. Heat-tolerant varieties utilize various mechanisms, including the expression of heat shock proteins, membrane integrity, and targeted gene expression that protect them from abiotic stresses like heat. We observed a strong negative correlation between the weighted multi-index selection score for best-performing potato genotypes under heat stress (based on yield, tuber minerals, and external and internal tubers defects) and relative leaf electrolyte conductivity and a positive correlation with total internal and external tuber defects. Also, our estimation of sHSP18 expression in leaves using Western blotting analysis showed that the heat-tolerant Vanguard Russet variety had significantly higher (threefold) expression than the heat-sensitive Russet Burbank, indicating HSP18 as a possible heat tolerance marker in potatoes. Heat tolerance at the plant level can be different than heat tolerance at the tuber level. Tubers are modified stems, and membrane stability at the leaf level could transfer to leaf membrane stability at the tuber level. Our study shows that the modified electrolyte leakage assay technique could be considered a fast and reliable technique for screening a large panel of diverse potato genotypes for heat tolerance and aid in identifying tentative heat-tolerant clones. The heat-tolerant clones evaluated in this study were confirmed as heat-tolerant based on having high yield and low external and internal defects under high-temperature conditions. Variations in field growing conditions (particularly environmental factors) should affect leaf REC values. The early screening of breeding clones under controlled high-temperature conditions (stringent 50°C level), for leaf membrane REC should be followed by field screening under stressful high-temperature conditions, using clones with previously detected low leaf REC values to confirm if clones with low leaf REC retain high marketable yields (high yields and low levels of external and internal tuber defects).

## Figures and Tables

**Figure 1 plants-13-02175-f001:**
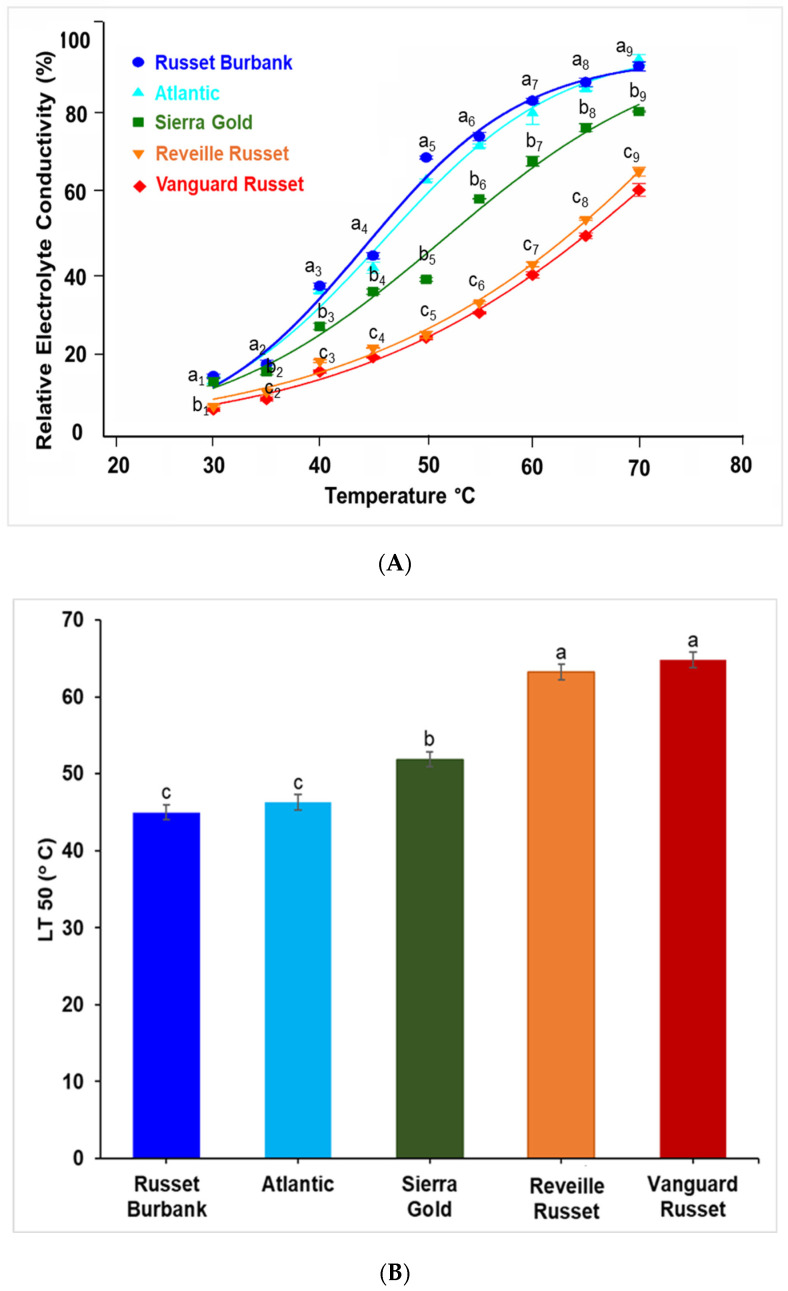
Leaf membrane relative electrolyte conductivity (REC) of Russet Burbank, Atlantic, Sierra Gold^TM^, Vanguard Russet, and Reveille Russet in the initial experiment: (**A**) changes in REC with increasing temperatures (30–70°C); (**B**) temperatures at which 50% of the leaf electrolytes leaked (LT50). Values with the same letter were not significantly different at *p* ≤ 0.05.

**Figure 2 plants-13-02175-f002:**
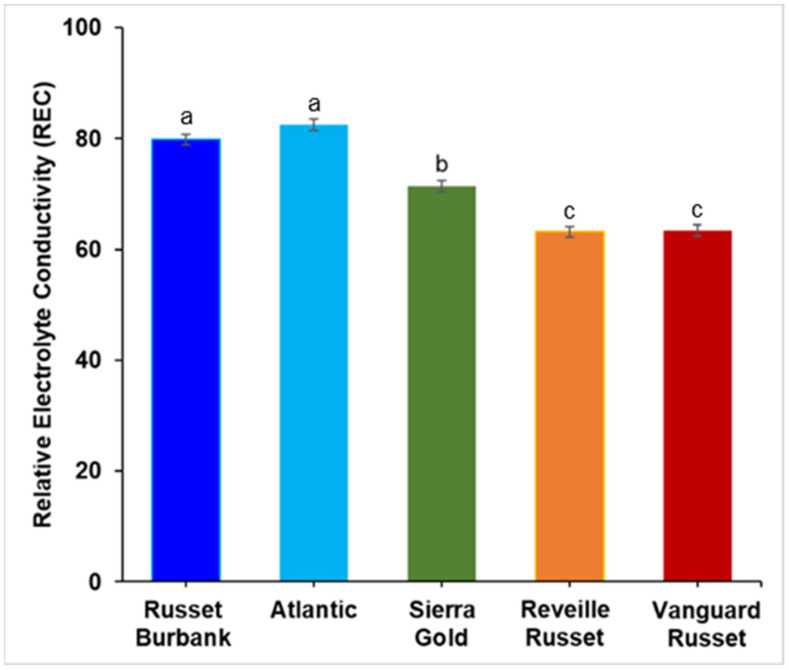
Leaf membrane relative electrolyte conductivity (REC) of Russet Burbank, Atlantic, Sierra Gold^TM^, Vanguard Russet, and Reveille Russet at 50°C in the main experiment. Values with the same letter were not significantly different at *p* ≤ 0.05.

**Figure 3 plants-13-02175-f003:**
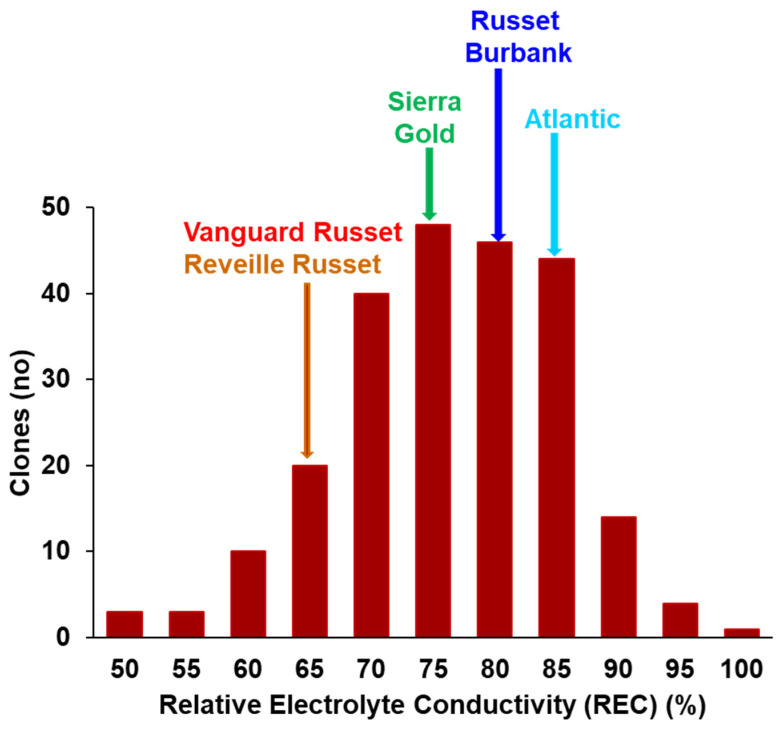
The overall distribution of 215 potato clones (advanced clones from the Texas A&M Potato Program and commercial varieties), based on leaf membrane relative leaf electrolyte conductivity (REC) measured from leaf disks exposed to four hours of heat stress in the water bath at 50°C.

**Figure 4 plants-13-02175-f004:**
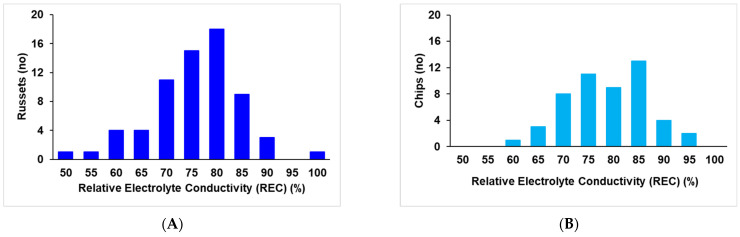
Distribution of 215 potato clones (advanced clones from the Texas A&M Potato Program and commercial varieties) according to individual market groups, namely (**A**) russets, (**B**) chips, (**C**) yellows, (**D**) reds, and (**E**) purples, based on relative leaf electrolyte conductivity (REC) measured from leaf tissues exposed to four hours of heat stress in the water bath at 50°C.

**Figure 5 plants-13-02175-f005:**
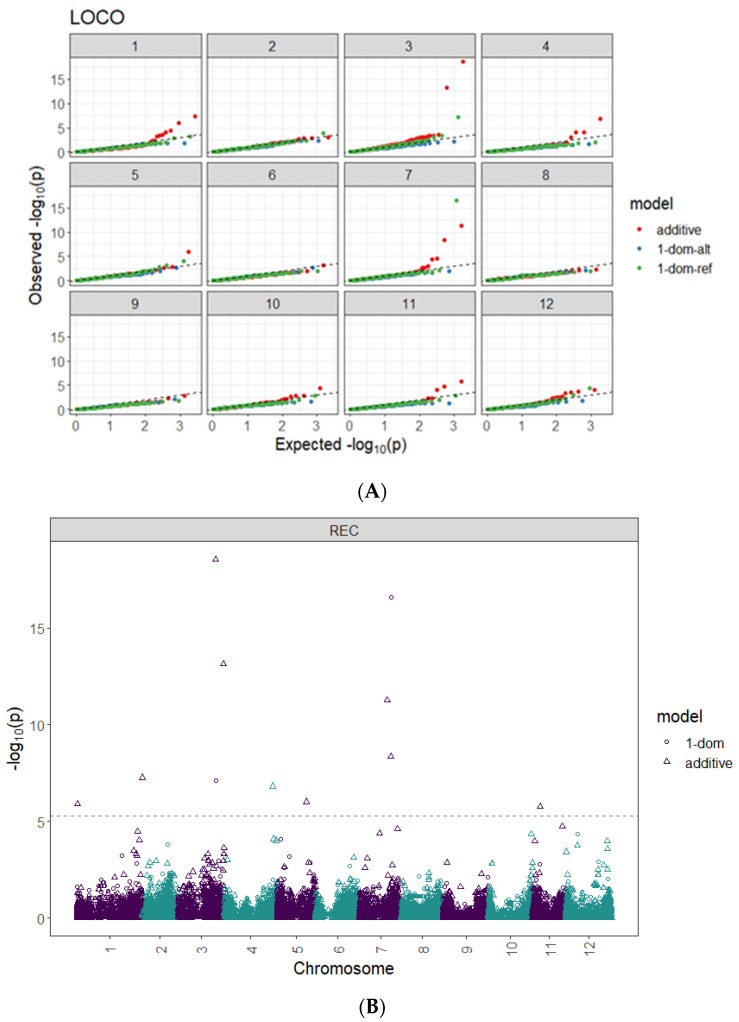
QQ plots of leaf relative electrolyte conductivity (REC) values (**A**) and Manhattan plot showing −log10 (*p*) values corresponding to SNPs across the 12 potato chromosomes for relative electrolyte conductivity using additive and dominant models (**B**). Genome-wide association studies (GWASs) were based on 215 potato clones evaluated for relative electrolyte conductivity (REC) after four hours of incubating leaf tissues in a water bath at 50°C. The Bonferroni threshold was 5.30 for the additive, 4.9 for 1-dom-alt, and 5.1 for the 1-dom-ref models.

**Figure 6 plants-13-02175-f006:**
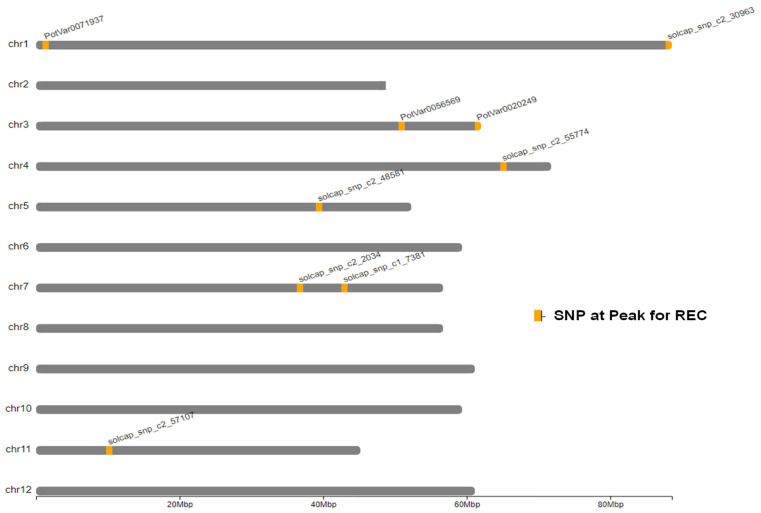
Location of significant QTLs on potato chromosomes for leaf membrane relative electrolyte conductivity (REC) based on evaluations of 215 potato genotypes. ChromoMap (Version 4.1.1) was used to develop the graph.

**Figure 7 plants-13-02175-f007:**
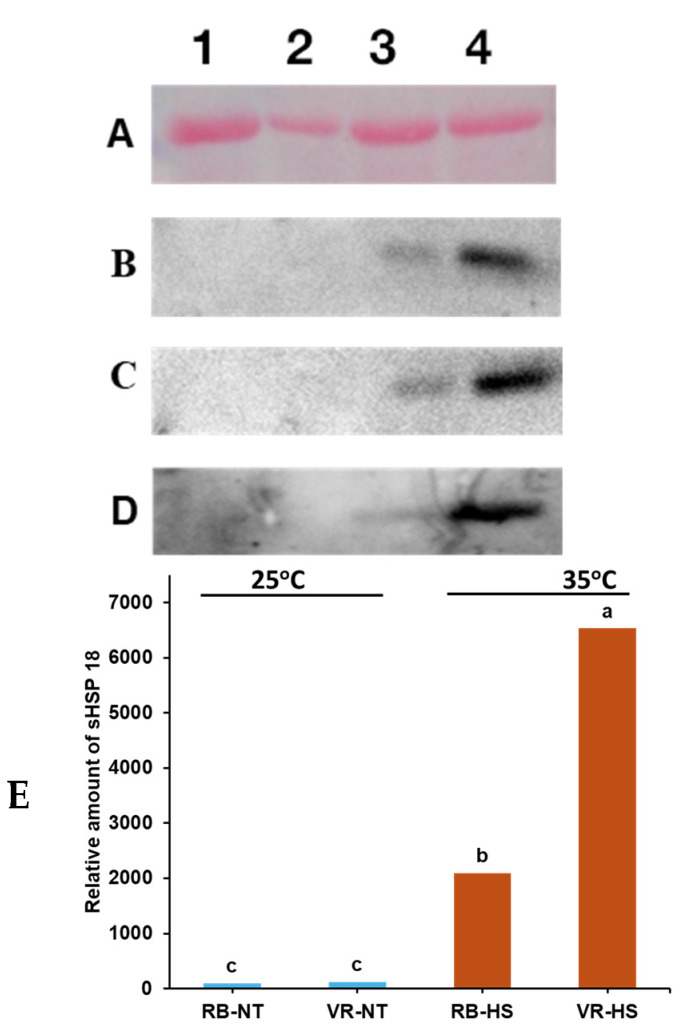
Heat stress-induced accumulation of cytosolic sHSP18 in (1) Russet Burbank—control, 25°C, RB-NT; (2) Vanguard Russet—control, 25°C; VR-NT, (3) Russet Burbank—heat stress, 35°C, RB-HS; and (4) Vanguard Russet—heat stress, 35°C, VR-HS grown under controlled growth chamber conditions: (**A**) total protein before Western blotting; (**B**–**D**) expression of sHSP 18 in Exp 1–3; (**E**) estimation of sHSP 18 based on band volume of Exp. 1. Expression of HSP18 was examined by Western blotting analysis in leaves of plants exposed to heat stress (35°C for 18 h) and control plants (25°C for 18 h). The blots were probed with the anti-*Arabidopsis thaliana* HSP17.6-CI polyclonal antibody. An equal amount of protein (20 µg) was loaded in each lane. The relative levels of HSP18 were estimated by determining band density after probing using ImageJ software (ver. 5.2, Molecular Dynamics, Sunnyvale, CA, USA). Values followed by the same letter in a column are not statistically different at *p* ≤ 0.05. Figure 8 summarizes the results obtained from electrolyte leakage assay, correlation analysis, and Western blotting in a schematic diagram.

**Figure 8 plants-13-02175-f008:**
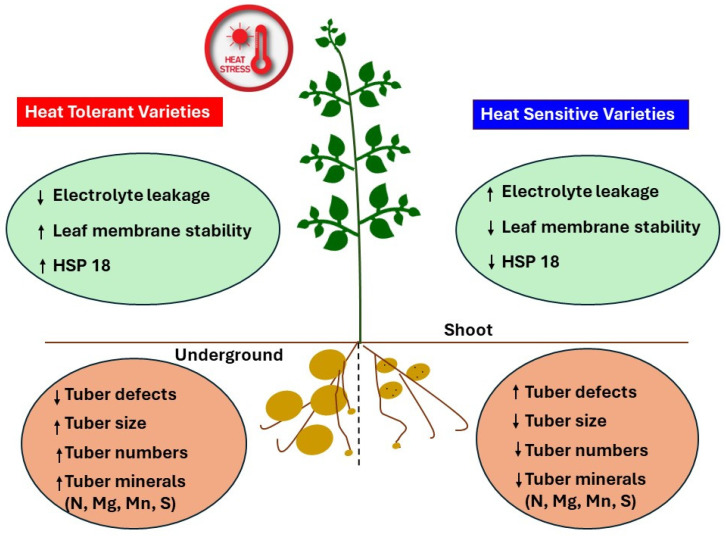
Under high temperatures, heat-tolerant potato varieties exhibited lower leaf electrolyte leakage, better membrane stability, and higher expression of small heat shock protein 18 (sHSP 18) compared to heat-sensitive varieties. Correlation analysis revealed that heat-sensitive varieties had higher tuber external and internal defect rates, lower yields, and reduced levels of essential minerals such as nitrogen (N), magnesium (Mg), manganese (Mn), and sulfur (S). {↑ = higher; ↓ = lower}.

**Table 1 plants-13-02175-t001:** Correlation probability and significance value for relative leaf electrolyte conductivity (REC) and computed Z-score values for total yield, percent tubers with external defects, percent tubers with internal defects, and weighted multi-trait index selection (WMIS) and total yield, yield components, overall external and internal defects and dissected external defects and internal defects of 215 potato genotypes evaluated in two locations in Texas (Dalhart and Springlake) in 2019, 2020, and 2021 (based on Gautam et al. [24].

Traits	Correlation to REC (%)	Probability	Sig. Level
Total Yield	−0.05	0.5710	ns
Yield without culls	−0.09	0.3200	ns
Culls	0.25	0.0042	**
% Tubers with knobs	0.33	0.0001	***
% Tubers with growth cracks	0.12	0.1830	ns
% Tubers with external defects	0.35	0.0001	***
% Tubers with hollow heart	0.15	0.0810	ns
% Tubers with vascular discoloration	0.12	0.1640	ns
% Tubers with internal heat necrosis	0.27	0.0070	**
% Tubers with internal defects	0.23	0.0040	**
Total Yield (Z)	−0.18	0.0380	*
% Tubers with external defects (Z)	0.40	0.0001	***
% Tubers with internal defects (Z)	0.34	0.0001	***
Weighted multi-index selection (Z)	−0.64	0.0001	***

ns = non-significant, * = *p* < 0.05, ** = *p* < 0.01, and *** = *p* < 0.001.

**Table 2 plants-13-02175-t002:** Correlation probability and significance value for relative leaf electrolyte conductivity (REC) and computed Z-score values for total tuber minerals and weighted multi-trait index selection (WMIS) of potato genotypes evaluated in two locations in Texas (Dalhart and Springlake) in 2019 and 2020 (based on Pandey et al. [62]).

Minerals	Correlation to REC (%)	Probability	Sig. Level
Nitrogen	−0.21	0.0160	*
Phosphorus	−0.17	0.0550	ns
Potassium	−0.08	0.3800	ns
Calcium	−0.10	0.2830	ns
Magnesium	−0.25	0.0051	**
Sodium	−0.13	0.1541	ns
Zinc	−0.08	0.3492	ns
Iron	−0.15	0.0843	ns
Copper	0.07	0.4391	ns
Manganese	−0.22	0.0121	*
Sulfur	−0. 20	0.0230	*
Boron	−0.17	0.0630	ns
Weighted multi-index selection (Z)	−0.25	0.0041	**

ns = non-significant, * = *p* < 0.05, ** = *p* < 0.01.

**Table 3 plants-13-02175-t003:** Quantitative trait loci (QTLs) for relative leaf membrane electrolyte conductivity (REC): SNP at the peak of each QTL, chromosome (Chr.), the position of the peak, score 1 − log10(p), and percentage of phenotypic variation explained by each QTL (R^2^).

Model	SNP at QTL Peak	Chr	Peak Position (bp)	Score 1 − log10(p)	R^2^ (%)
Additive	PotVar0071937	1	1,158,384	5.9	2.60
Additive	solcap_snp_c2_30963	1	88,362,869	7.3	1.40
Additive	PotVar0056569	3	50,880,446	18.5	4.30
Additive	PotVar0020249	3	61,550,994	13.1	1.50
Additive	solcap_snp_c2_55774	4	65,972,478	6.8	0.01
Additive	solcap_snp_c2_48581	5	39,349,874	6.0	1.50
Additive	solcap_snp_c2_2034	7	37,157,836	11.3	3.90
Additive	solcap_snp_c1_7381	7	42,659,912	8.4	1.20
Additive	solcap_snp_c2_57107	11	10,505,722	5.8	0.05
1-dom-ref	PotVar0056569	3	50,880,446	7.1	8.20
1-dom-ref	solcap_snp_c1_7381	7	42,659,912	16.6	13.80

## Data Availability

The raw data supporting the conclusions of this article will be made available by the authors, without undue reservation.

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
