# Peer review of "Leaf Membrane Stability under High Temperatures as an Indicator of Heat Tolerance in Potatoes and Genome-Wide Association Studies to Understand the Underlying Genetics"

_plants, 2024, doi:10.3390/plants13162175_

Round 1
Reviewer 1 Report
Comments and Suggestions for Authors
The research article titled "Leaf membrane stability under high temperature as an indicator of heat tolerance in potatoes and genome-wide association studies to understand the underlying genetics" addresses a critical challenge in modern agriculture: the impact of rising global temperatures on crop productivity and quality. This study explores the relationship between leaf membrane stability, as measured by relative electrolyte conductivity (REC), and heat tolerance in potato genotypes. Utilizing a combination of electrolyte leakage assays, correlation analyses, and genome-wide association studies (GWAS), the research provides insights into the genetic basis of heat tolerance and proposes potential markers for breeding programs. In order to improve the present study, some essential modifications have to be fixed before it proceeds, and decisive action can be taken. All the comments and remarks are given below.
Abstract
The abstract effectively summarizes the research. Significant findings are presented concisely, such as the variation in REC among genotypes, the correlation of REC with tuber traits, and the identification of QTLs.
Introduction
The introduction effectively establishes the context by discussing the general impact of high temperatures on plant physiology and cellular processes.
The background is well-presented, but it could benefit from a more focused discussion on potatoes specifically, rather than general plant responses.
The introduction is too lengthy, some sections could be more concise to maintain focus on potatoes.
Some references are quite dated (such as, Raison et al., 1980, Quinn, 1988, Ismail & Hall, 1999, Ashraf et al., 1994); where possible, include more recent studies to reflect current understanding and methodologies in the field.
Material and Methods
The Materials and Methods section is comprehensive, detailed, and well-organized. Each step is described clearly, allowing for reproducibility.
Please mention the rationale behind choosing specific temperatures and durations for heat treatments, to provide better context.
Results
The Results section is comprehensive, well-organized, and clearly presented. The use of figures and tables enhances the clarity and understanding of the results.
Please include a brief discussion on why the incubation time was increased to 4 hours in the main study compared to 30 minutes in the initial study.
Authors should include a brief discussion on the potential biological significance of the identified QTLs and their associated genes, this would enhance the interpretation of the results.
Discussion
The discussion is well-organized, thorough, and effectively synthesizes the study's findings with existing knowledge. It highlights the practical implications for breeding programs and suggests future research directions.
The correlation between REC and tuber quality traits is discussed, but the biological implications of these correlations need deeper exploration. For instance, why might genotypes with higher REC have more defects?
The discussion on sHSP18 is clear and well-supported by previous studies. However, integrating a discussion on how these findings could be used in breeding programs (e.g., marker-assisted selection) would enhance practical relevance.
Conclusion
The conclusions effectively tie back to the study’s objectives, providing a coherent wrap-up.
Authors should elaborate on the physiological mechanisms by which lower REC translates to better heat tolerance. This could involve discussing the specific roles of identified QTLs and associated genes in maintaining membrane integrity under stress.
Authors should also acknowledge any limitations of the study, such as the potential variability in field conditions that might affect the transferability of greenhouse results to actual growing conditions.
Overall, the manuscript makes a significant contribution to the understanding of heat tolerance in potatoes. By addressing the points above, the clarity, depth, and impact of the study can be further enhanced, making it a more comprehensive and critical piece of research.
Author Response
Thank you for taking the time to review our manuscript and providing constructive feedback to help us improve our work.
The review has been attached as a Word document.

Reviewer 2 Report
Comments and Suggestions for Authors
This manuscript claims that leaf membrane stability can be used as an indicator of heat tolerance in potato, however there are still some issues that need to be addressed before the manuscript can be accepted.
1. There are many methods for evaluating heat resistance. Why choose cell membrane stability? Is it convenient to measure? Is it easy to observe? What is the practical significance?
2. In the background, there is a lack of current problems, thus failing to highlight the importance of the research.
3. The title of the result should be what was obtained rather than what was done.
4. The manuscript contains rich data, please add causal analysis to clarify the relationship between leaf membrane homeostasis and other factors involved in the manuscript.
5. The discussion lacks the problems of this study.
6. Figures in manuscripts should wait until optimized.
Author Response

(The authors gave the same response as above.)

Reviewer 3 Report
Comments and Suggestions for Authors
The manuscript was well presented and well documented. Although not highly
innovative, it is a solid piece of research.
Author Response
Reviewer 3
The manuscript was well presented and well documented. Although not highly innovative, it is a solid piece of research.
Response:
Thank you for your positive feedback. Our study focused on developing a practical and applicable method to screen for heat tolerance in potatoes. Additionally, employing genome-wide association studies allowed us to explore the genetic link between electrolyte leakage and heat tolerance, an aspect not previously reported in potatoes.